# High-Sensitivity Determination of Nutrient Elements in *Panax notoginseng* by Laser-induced Breakdown Spectroscopy and Chemometric Methods

**DOI:** 10.3390/molecules24081525

**Published:** 2019-04-18

**Authors:** Tingting Shen, Weijiao Li, Xi Zhang, Wenwen Kong, Fei Liu, Wei Wang, Jiyu Peng

**Affiliations:** 1College of Biosystems Engineering and Food Science, Zhejiang University, 866 Yuhangtang Road, Hangzhou 310058, China; shentingtingstt@163.com (T.S.); 15236193955@163.com (W.W.); jypeng@zju.edu.cn (J.P.); 2Chinese Materia Medica, Yunnan University of Chinese Medicine, Kunming 650500, China; liweijiao163@163.com (W.L.); zhangxi1030@hotmail.com (X.Z.); 3School of Information Engineering, Zhejiang A & F University, Hangzhou 311300, China; wwkong16@zafu.edu.cn; 4Key Laboratory of Spectroscopy Sensing, Ministry of Agriculture and Rural Affairs, Hangzhou 310058, China

**Keywords:** *Panax notoginseng*, nutrient elements, laser-induced breakdown spectroscopy, least absolute shrinkage and selection operator, matrix effect, traditional Chinese medicine

## Abstract

High-accuracy and fast detection of nutritive elements in traditional Chinese medicine *Panax notoginseng* (PN) is beneficial for providing useful assessment of the healthy alimentation and pharmaceutical value of PN herbs. Laser-induced breakdown spectroscopy (LIBS) was applied for high-accuracy and fast quantitative detection of six nutritive elements in PN samples from eight producing areas. More than 20,000 LIBS spectral variables were obtained to show elemental differences in PN samples. Univariate and multivariate calibrations were used to analyze the quantitative relationship between spectral variables and elements. Multivariate calibration based on full spectra and selected variables by the least absolute shrinkage and selection operator (Lasso) weights was used to compare the prediction ability of the partial least-squares regression (PLS), least-squares support vector machines (LS-SVM), and Lasso models. More than 90 emission lines for elements in PN were found and located. Univariate analysis was negatively interfered by matrix effects. For potassium, calcium, magnesium, zinc, and boron, LS-SVM models based on the selected variables obtained the best prediction performance with *R_p_* values of 0.9546, 0.9176, 0.9412, 0.9665, and 0.9569 and root mean squared error of prediction (RMSEP) of 0.7704 mg/g, 0.0712 mg/g, 0.1000 mg/g, 0.0012 mg/g, and 0.0008 mg/g, respectively. For iron, the Lasso model based on full spectra obtained the best result with an *R_p_* value of 0.9348 and RMSEP of 0.0726 mg/g. The results indicated that the LIBS technique coupled with proper multivariate chemometrics could be an accurate and fast method in the determination of PN nutritive elements for traditional Chinese medicine management and pharmaceutical analysis.

## 1. Introduction

*Panax notoginseng* (commonly known as Sanqi or Tianqi) is a valuable Chinese herbal medicine that is in clinical trials or in clinical practice [1,2]. The U.S. Dietary Supplement Health and Education Act advocated *Panax notoginseng* (PN) as a dietary supplement in 1994 [3]. Modern research has shown that PN has biological activities in many aspects such as the blood system, cardiovascular system, nervous system, immune system, etc. [4]. Especially for heart and cerebrovascular diseases, PN can reduce myocardial oxygen consumption, improve myocardial ischemia, protect experimental cerebral ischemia, and is anti-heart rhythm [5,6]. PN also has good health functions such as lowering blood fat, lowering blood pressure, anti-shock, etc. [6].

Good health functions and medicinal value have increased the demand for PN in many countries. China is a major exporter of PN, and most of the PN comes from Yunnan Province [7]. The complexity of Yunnan’s terrain makes PN’s quality different in different producing areas [8]. The most obvious difference is nutrient elements in *Panax notoginseng*, because elemental content is directly related to the growth environment of the herbal medicine, as roots assimilate elements readily from the soil [9]. Nutritive elements in herbal medicine not only provide necessary nutrient elements for the human body [10], but also play significant roles in the formation of functional constituents such as Fe and Zn, which are the active centers of some metalloenzymes [11,12,13]. Therefore, it is important to analyze the nutritive elements to control the quality of traditional Chinese medicine and that serve as the basis for pharmacological research.

Due to the importance of nutritive elements present in traditional Chinese medicine, common detection technologies including inductively-coupled plasma optical emission spectrometry (ICP-OES) [14], inductively-coupled plasma-mass spectrometry (ICP-MS) [15], and atomic absorption spectrometry (AAS) [16] were used to determine nutritive elements to analyze the value of herbs. Although the techniques above could obtain accurate results for elements, the samples must undergo complex and time-consuming pretreatments such as high temperature and high acid digestion [17]. These defects limit on-line and massive detection of nutritive elements in PN from different areas.

Laser-induced breakdown spectroscopy (LIBS) is a novel atomic technique that exhibits the advantages of fast analytical speed, multi-element analysis, minimal sample preparation, and high efficiency [18,19]. These attractive features have attracted researchers to provide laboratory analysis to explore the potential for LIBS to be a polyvalent monitoring and analysis tool for elements’ detection [20], and especially nutritive elements in plant materials [21]. Braga et al. applied LIBS to analyze micronutrients in pellets of plant materials, and the results proved that LIBS technique combined with partial least-squares (PLS) is robust for elements determination [22]. Carvalho et al. compared femtosecond LIBS and nanosecond LIBS to profile nutrients in plant materials, and the result showed that the content values predicted by nanosecond LIBS multivariate modeling exhibit better agreement with reference mass fractions [23]. Those studies indicated that the LIBS technique has the potential to detect nutritive elements, and all mention that the “matrix effect” could affect the performance of the detection model. Matrix effects are related to the entire ablation/detection process and the ablation differences rooted in the chemical composition and physical properties of samples [24]. The composition of PN is complex, and the matrix effect is inevitable. He et al. detected soil nutrient elements potassium (K), calcium (Ca), magnesium (Mg), iron (Fe), manganese (Mn), and sodium (Na) based on LIBS and found that the PLS model was the most suitable for quantitative analysis of elements [25]. The PLS method proved to be a robust model, mainly because PLS could reduce multicollinearity among variables and mitigate interference from the matrix effect. Other studies have also confirmed that SVM and Lasso are also good methods to build excellent regression models such as SVM models for quantitative analysis of Cr and Ni in iron alloys [26] and Lasso models for major element analysis of rocks [27].

*Panax notoginseng* in different producing areas has complex elemental compositions and structural differences. To the best of our knowledge, the quantitative detection of PN nutrient elements based on LIBS combined with chemometrics has not been investigated. The objective of this work is to explore fast and accurate quantitative detection of nutritive elements K, Ca, Mg, Fe, zinc, and boron in PN samples from eight producing areas by laser-induced breakdown spectroscopy. LIBS spectra were used to analyze element differences and elemental fingerprinting in PN samples from different regions. Univariate calibration and multivariate calibration were employed to establish the calibration models for quantitative analysis of elements. Especially, PLS, SVM, and Lasso were compared to eliminate matrix effect interference and improve the sensitivity and stability of LIBS detection.

## 2. Materials and Methods

### 2.1. Panax notoginseng Samples

All PN samples were obtained from 8 areas of Yunnan province in China, namely Xichou (1), Yongde (2), Malipo (3), Mile (4), Gejiu (5), Gengma (6), Shizong (7), and Qiubei (8). All samples were purchased in 2017 harvest seasons. Thirteen pieces of PN were sampled from the PN set of each origin randomly, and a total of 104 PN samples was collected for further analysis. Each piece of PN was dried in an oven at 40 °C for approximately 3 h and was ground into powder by a tissue milling machine for consistent measurement. Two hundred milligrams of each PN powder were placed into a squared die set and pressed into a pellet with 700 MPa of pressure for 30 s. A total of 104 pellets was obtained, each of which was square with a 10-mm length for each side. 

### 2.2. Spectral Acquisition

A self-assembled LIBS setup was used to realize LIBS spectral acquisition [28]. Laser pulses at 532 nm with maximum energy of 200 mJ and an 8-ns pulse width were generated by a Q-switched Nd:YAG pulse laser (Vlite 200, Beamtech, Beijing, China). The green laser passed through our self-made optical system and focused on 2 mm below one PN pellet’s surface through a plano-convex lens (f = 100 mm). The laser ablated PN and generated a high temperature plasma, which contained atoms, ions, electrons, and molecules. During out diffusion of plasma, each element in the plasma ionized to form continuous spectral lines. The information of elements was collected by a light collector and received by the spectrometer (ME5000, Andor Technology, Belfast, U.K.) combined with an ICCD camera (DH334T-18F-03, Andor Technology, Belfast, U.K.). The spectra between 230.77 and 883.24 nm with high resolution (λ/∆λ = 5000) were collected. A delay generator (DG645, Stanford Research Systems, Sunnyvale, CA, USA) was used to regulate the delay time between the ICCD camera and laser Q-switch that controlled the laser generation. The optimal experimental parameters were optimized with a laser energy of 60 mJ, a delay time of 2.827 μs, and a gate width of 20 μs. Each PN pellet was placed on a sample stage of an x-y-z motorized positioning system, which controlled the laser ablation path with 4 × 4 array. Therefore, 4 × 4 craters appeared on the pellet surface, and each crater had 5-times accumulation of laser pulses. The spectrum for each sample was recorded by the average of the 80 spectra (4 × 4 × 5) to reduce fluctuation between the laser point-to-point. The interval distance of each hit was 2 mm.

### 2.3. Reference Method for Nutrient Elements’ Content Determination

The reference method for detecting nutrient elements K, Ca, Mg, Fe, zinc (Zn), and boron (B) content in PN samples mainly relied on an inductively-coupled plasma optical emission spectrometer (ICP-OES) [29]. Before the determination by ICP-OES, the PN samples needed to experience microwave digestion and acid discharge pretreatment. Each pellet after LIBS acquisition was weighed and placed into modified polytetrafluoroethylene vessels with 5 mL of 65% HNO_3_ and 1 mL of 30% H_2_O_2_ for microwave digestion at 185 °C. Then, modified polytetrafluoroethylene vessels with digested liquid were placed in a 165 °C furnace to discharge the acid till a drop of fuming digested liquid remained. The acid elimination pretreatment happened in the fuming cupboard. The least digested liquid was diluted to a volume of 30 mL with high-purity water by the weighing method. The final dilution was applied to detect elemental content by ICP-OES.

### 2.4. Data Analysis

#### 2.4.1. Data Preprocessing

For remitting systematic and random errors during the experiment, wavelet transform (WT) was used to preprocess the raw spectral. As an efficient denoising method, WT uses a set of wavelet basis functions to remove invalid information (noise) and reserve valid sharp peaks or spikes [30]. Daubechies 8 with the decomposition scale of 3 was selected in our paper. After WT preprocessing, the spectral of 104 PN samples were sorted from the lowest to highest according to K, Ca, Mg, Fe, Zn, and B content values (by ICO-OES), respectively. The Kennard–Stone (KS) algorithm [29], which could avoid bias in sample selection, was used to split 104 PN samples into a calibration set and a validation set. Seventy two PN samples were selected for the calibration set, while the remaining 32 samples formed the prediction set. Univariate analysis and multivariate analysis were initially done using PN samples in the calibration set. 

#### 2.4.2. Multivariate Analysis Methods

Multivariate analysis employed PLS, SVM, and Lasso to establish calibration models, and the parameter selection of models was done using cross-validation. Lasso was used to select effective variables to improve multivariate analysis performance.

Partial least-squares regression (PLS) is an established analytical tool for relating multivariate data analysis [31]. A PLS model expresses the relation between LIBS spectral information X and the content of element Y in PN samples. Spectral matrix X and content matrix Y were decomposed, and the invalid information in the two matrixes was handled simultaneously. Principal component information was calculated after matrix decomposition. When calculating the principal component, PLS considered larger variance of the principal component to extract more useful information and also made the principal component (latent variables (LVs)) and element concentration Y more relevant to maximize the linear relationship between spectral variables and concentrations [32,33]. The five-fold cross-validation procedure was used, and the number of LVs was selected when the first minimum or the knee point for the root mean squared error of cross-validation (RMSECV) in calibration set vs. LVs curve was obtained. 

Least-squares support vector machine (LS-SVM) is an improvement of the standard SVM based on the structural risk minimization (SRM) approach proposed by Vapnik et al. [34]. LS-SVM used the least-squares linear system as the loss function and applied a set of linear equations to replace the complicated quadratic programming method, which was adopted by the standard SVM. LS-SVM reduces the computational complexity and solution speeds and improves generalization ability of model [35]. Compared with PLS, LS-SVM could not only solve linear relation, but also nonlinear regression problems. The radial basis function (RBF) kernel function and five-fold cross-validation were utilized to establish the LS-SVM calibration model. The penalty parameters (c) and kernel function parameters (g) of LS-SVM were optimized by a grid-search procedure in the range of 103–1010, and the best c and g were determined with the minimal value of RMSECV.

The least absolute shrinkage and selection operator (Lasso) is a penalized shrunken regression method [36]. Lasso is also a dimensionality reduction method for both linear and nonlinear cases. The penalty method L1 was introduced when spectral matrix X did not belong to column full rank. Lasso selected variables from spectral data according to the penalty method L1 [27]. The penalty method L1 compressed the original coefficients β, and some original small coefficients were directly compressed to 0. The variables corresponding to these β = 0 were regarded as non-significant variables to be discarded directly. The penalty method took the value when the penalty likelihood function was the smallest as the estimated value of the regression coefficient [27,37]. Five-fold cross-validation was also applied to establish the Lasso calibration model and confirm the best model parameter the boundary value t, which is analogous to LVs for PLS and the number of non-zero β, which is expressed as either t in the optimization equation or the number of steps in the step-wise procedure.

#### 2.4.3. Performance Evaluation

The correlation coefficient (R), RMSECV, and root mean squared error of the prediction set (RMSEP) were used to evaluate the performance of the quantitative models for nutritive elements’ content detection [38]. Correlation coefficient (R) means correlation between target element content obtained by ICP-OES and target element content detected by LIBS. *R_c_* is correlation coefficient of the calibration set and *R_p_* is the correlation coefficient of the prediction set. The closer the R value is to 1 with the smaller root mean squared error, the better the performance and detectability of LIBS variables and calibration models.

#### 2.4.4. Software Tools

LIBS spectra acquisition was carried out by Andor SOLIS for Imaging (v4.26, Andor Technology, Belfast, U.K.). Data analysis was executed by MATLAB R2017a (The MathWorks, Inc., Natick, MA, USA). Origin Pro 2015 (Origin Lab Corporation, Northampton, MA, USA) was applied for graphs’ design.

## 3. Results

### 3.1. Nutritive Elements Content of Panax Notoginseng

Nutritive elements K, Ca, Mg, Fe, Zn, and B content of eight producing areas detected by ICP-OES are shown in Table 1. For PN samples from all regions, the content of B and Zn was much smaller than that of Fe, which also belonged to the microelement. Commonly belonging to nutritive elements, the content of K was nearly ten-times that of Ca and Mg in all PN samples. The K, Ca, Mg, Fe, and Zn content of PN in Malipo (Group 3) was higher than the other regions, and the B content was second only to the PN in Xichou (Group 1), which indicated that the quality of PN in Malipo is best. Compared with other regions, the content of those elements in PN samples from Gejiu (Group 5) and Gengma (Group 6) was relatively small. The content of these nutritive elements could reflect the quality of PN goods from different producing areas. Undoubtedly, the method of rapid and simultaneous detection of elements content is conducive to the regulation of the PN commodity market.

### 3.2. LIBS Spectra Analysis

The average spectrum of each area in the range of 230.77–883.24 nm is shown in Figure 1. The LIBS spectra of PN samples from the eight areas were observed to have a similar tendency, which signified that PN in different producing areas had the same element species and similar matrix compositions. Molecule bands CN and H and atomic line O I, which commonly appeared in the LIBS spectra of organic samples, can be observed in Figure 1. The emission lines of K, Ca, Mg, and Fe in eight areas of PN samples are also obvious in Figure 2 with different intensity values such as PN in Gengma (Group 6) having the lowest Mg emission line and PN in Malipo (Group 3) having the highest Mg emission line, which is consistent with the results in Table 1. The intensity difference of element emission lines indicated that different habitats significantly changed the content of PN and formed a specific proportion of elements.

Based on the Kurucz database and the National Institute of Standards and Technology (NIST) Atomic Spectra Database (ASD), more than 90 emission lines whose intensity values were above 1000 a.u. were verified identities, and the elements are shown in Table 2. Nutritive element Ca had 35 emission lines, and Ca II 393.37 nm had the maximum intensity value of all element emission lines in the PN samples. The emission line for carbon (C) without the highest intensity value may be due to the relatively long delay time, since the intensity of emission lines changed quickly within the time [18]. The observed emission lines for oxygen and nitrogen may come from the common contribution of the PN sample and air. Table 2 presents the capability of LIBS technique to detect different elements from PN samples. 

### 3.3. Univariate Analysis

Univariate analysis is a calibration curve method that reflects that the intensity of emission line is proportional to the content of the target element in samples [39]. In common LIBS quantitative analysis, sensitive emission lines of the target element without self-absorption and overlapping peaks are the preferred choice for univariate analysis [18]. According to the NIST database, the sensitive emission lines K I 766.49 nm, K I 769.90 nm, Ca II 393.37 nm, Ca II 396.85 nm, Ca I 422.67 nm, Mg I 517.27 nm, Mg I 518.36 nm, Fe I 373.71 nm, and Fe I 371.99 nm were selected to build univariate analysis models for *Panax notoginseng*. As shown in Figure 2, the emission lines of these nutritive elements are smooth without interference peaks. The peak intensity of K, Ca, Mg, and Fe in Groups 2, 5, and 6 was significantly lower than other groups, which is consistent with the difference in element content in Table 1. Because of the low content, the sensitive emission lines of B and Zn were difficult to discriminate for univariate analysis.

The univariable calibration and prediction results of the above selected emission lines of nutritive elements are shown in Table 3. For univariate analysis of K content, K I 769.90 nm performed better than K I 766.49 nm, with *R_c_* of 0.8413, RMSECV of 1.370 mg/g in calibration, and *R_p_* of 0.7836, RMSCP of 1.610 mg/g in prediction. The results of Ca I 422.67 nm, Mg I 517.27 nm, and Fe I 371.99 nm were all better than the other same element emission lines. Overall, the univariate analysis model for Fe I 371.99 nm performed best with *R_c_* of 0.8944, RMSECV of 0.082 mg/g in calibration and *R_p_* of 0.8577, RMSCP of 0.097 mg/g in prediction. However, the performance of univariate analysis in Table 2 is not sufficient to generate a robust and accurate predictive model for the content of K, Ca, Mg, and Fe in PN samples.

### 3.4. Multivariate Analysis

Multivariate analysis attempted to analyze the relationship between more variables of LIBS spectra and nutrient elements K, Ca, Mg, Fe, Zn, and B content obtained by ICP-OES. Full spectra or several selected variables combined with chemometric methods PLS, LS-SVM and Lasso were used to establish calibration models for the objective nutritive elements content.

#### 3.4.1. Modeling Using Full Spectra

The results for multivariate analysis based on full spectra from 230.77 nm–883.24 nm are shown in Table 4. For K and Ca, the multivariate analysis by Lasso and PLS achieved good performance with *R_p_* higher than 0.9500. For Mg and Fe, the multivariate analysis based on full spectra by Lasso achieved the best performance with *R_p_* values of 0.9207 and 0.9348 and RMSEP of 0.7740 mg/g and 0.0722 mg/g, respectively. For Zn and B, PLS models obtained the best performance with *R_p_* values of 0.9460 and 0.9475 and RMSEP of 0.0016 mg/g and 0.0010 mg/g, respectively. 

For all the target elements in the PN samples, PLS and Lasso analysis performed well in multivariate analysis based on LIBS full spectra, which may be because PLS and Lasso were more suitable for solving multi-collinearity problems, making full use of the useful element information from the 22,036 variables for relationship fitting between LIBS and element content. LS-SVM performed well in calibration sets for all tested elements, but poor in prediction sets, which means over-fitting phenomenon occurred. A huge difference in the number of variables and the number of samples may cause this phenomenon.

#### 3.4.2. Modeling Using Selected Variables

During the Lasso modeling process, irrelevant or insignificant coefficients of corresponding variables were penalized to zero and discarded directly [40]. Therefore, relevant or significant variables could be selected by comparing regression coefficients (weights) for each variable. The weights plot of the Lasso models for the nutrient elements are shown in Figure 3.

In the case of K, the emission line at 769.89 nm showed strong correlation to the concentration. Ca II 393.36 nm and Ca II 396.84 showed strong correlation to the Ca concentration; Mg I 517.27 nm and Mg I 383.83 nm showed correlation to Mg concentration; Fe I 371.99 nm, Fe I 438.35 nm, and Fe I 537.15 nm showed strong correlation to Fe concentration; Zn I 330.25 nm, Zn I 334.50 nm, and Zn I 692.83 nm showed strong correlation to Zn concentration; B I 249.77 nm showed strong correlation to B concentration. It is clear that other multiple lines such as N, O, H, and Ca exist in the weight plots of each element (Figure 3), which is due to the complex matrix components of PN and random noise. There were high weights of multiple lines such as N and O in K, Ca, Mg, Zn, and B, which indicated that these elements contributed significantly to quantitative analysis of the target elements, and the interactions among all the elements in PN herbs could not be ignored. The target elements may combine with N and O to compose compounds such as aerobic compounds or amino acids. The multiple lines like N and O may also come from air ablation and reflect noise from environmental fluctuation. Compared with the spectrum of one PN sample (Figure 3d,h), we also found that background signals without emission lines were also contributing to the quantification study of nutrient elements such as the unmarked weight lines in the Lasso weights plot for Mg (Figure 3c). The background information with high weights may be related to the matrix effects of the PN samples.

The LIBS variables with non-zero weights values in Lasso models were selected for multivariate analysis by PLS, LS-SVM, and Lasso, and the results are shown in Table 5. The number of LIBS variables selected by Lasso for K, Ca, Mg, Fe, Zn, and B was 64, 73, 61, 66, 73, and 62, respectively. For K, Ca, Mg, Zn, and B content predictions, PL-SVM models based on the selected variables by Lasso achieved the best performance with *R_p_* values of 0.9546, 0.9176, 0.9412, 0.9665, and 0.9569 and RMSEP of 0.7704 mg/g, 0.0712 mg/g, 0.1000 mg/g, 0.0012 mg/g, and 0.0008 mg/g, respectively. For Fe content predictions, PLS models achieved the best performance with a *R_p_* value of 0.9169 and an RMSEP of 0.0724 mg/g. On the whole, LS-SVM models could effectively predict the content of elements, followed by PLS models. Lasso quantitative analysis based on the selected variables preformed worse than the other analysis methods based on the same variables and Lasso quantitative analysis based on full spectra with 22,036 variables. This is because Lasso quantitative analysis based on the selected variables may lose some valid variables after over-screening these variables by the penalty method *L_1_*.

## 4. Discussion

For better univariate analysis, the emission lines without self-absorption and overlapping peaks should be chosen (Figure 2). The spectral emission line intensity is related to the element content. For the lower content elements, the emission line is often too weak, which makes it difficult to distinguish the sensitive emission line and may cause misjudgment. Therefore, content prediction of Zn and B, which are trace elements with relatively low amounts, was performed with multivariate analysis. However, univariate analysis for the nutritive elements obtained poor results. The matrix effect in complex PN samples has a major responsibility for the poor results.

Matrix effects come from the complex plant tissue including differences in chemical compositions and physical properties of plant tissue such as hardness, roughness, porosity, and density [28]. Matrix effects also relate to optical and plasma properties that influence the ratio of a given emission line to the abundance of the element producing that line [27]. Table 2 shows more than 90 emission lines with intensity values being above 1000 a.u. The LIBS spectra contain not only sensitive emission lines of the target elements, but also a large amount of other elements’ information containing matrix effects’ information. Figure 3 also indicates that multiple emission lines of other correlated elements and background variables may contribute to the prediction of target elements. Univariate analysis only considered the intensity of the sensitive emission line and lose of information related to the matrix effects. Therefore, univariate analysis is not an appropriate way to detect the nutrient elements’ content in PN herbs.

Unlike univariate calibration, multivariate analysis performed well in Table 4 and Table 5 and had the ability to excavate more multi-variables, including not only the useful information from sensitive emission lines of object element, but also the complex information from matrix effects, continuous background, and shot-to-shot fluctuation of the laser [17]. 

The effective variables are the critical point of whether to choose univariate or multivariate analysis. Full spectra had 22,036 variables including effective variables and vast ineffective variables, which inevitably resulted in model complexity and instability. After Lasso was selected, the effective LIBS variables reduced from 22,036 to 64, 73, 61, 66, 73, and 62 for K, Ca, Mg, Fe, Zn, and B, respectively. The variable screening process could significantly reduce the number of variables in LIBS spectra for model input and mitigate noise or irrelevant information from background interference and matrix effects. The results of LS-SVM models based on the selected variables clearly demonstrated the merit of variable screening by Lasso. After the variable screening, the *R_p_* and RMSEP of the SVM models were greatly improved. Comparing Table 4 and Table 5, LS-SVM models based on the selected variables by Lasso weights obtained the best prediction performance for K, Ca, Mg, Zn, and B (*R_p_* = 0.9546, 0.9176, 0.9412, 0.9665, and 0.9569, respectively). For Fe content prediction, Lasso based on full spectra got the best result with *R_p_* = 0.9207. The fitting plots for the best results for K, Ca, Mg, Fe, Zn, and B are shown in Figure 4.

PLS yielded comparable results in terms of accuracy both based on full spectra and the selected variables by Lasso weights. Therefore, PLS models could simultaneously account for changes in all these variables in the condition of changing environments and infinitely-variable sample chemistries. For K, Ca, Mg, Fe, Zn, and B content detection of PN herbs, PLS may be better suited for this purpose. Lasso may be better for Mg and Fe content detection of PN herbs without more variable screening analysis, because the penalty method L1 selected variables from full spectral and the weights showed us the key emission lines used to quantify each element. The performance of LS-SVM models was limited by overfitting when confronted with full spectra. This may be a limitation of “large” LIBS variables and “small” sample sizes. Except Fe content analysis, the overfitting of LS-SVM models was alleviated after a reduction of more than 99.67% for variables by Lasso weights and showed the merit of the LS-SVM method dealing with the nonlinear relationship in the matrix effects and background interference. The LS-SVM models based on selected variables by Lasso were are the most suitable methods for prediction of K, Ca, Mg, Zn, and B content in PN herbs.

The quantitative analysis profiled the accuracy of LIBS combined with chemometrics, and our method also showed the ability of rapid detection for nutritive element content of PN herbs. Compared with the ICP-OES procedure, which needs more than a 150-min pretreatment containing weighting, adding other reagent, digesting, discharging acid, and diluting, the LIBS procedure needs less than 360 s, because the grinding and tableting process of a PN sample is less than 250 s and LISB collecting of information of the pellet needs about 60 s.

## 5. Conclusions

In this experiment, we demonstrated the rapid and accurate analysis of K, Ca, Mg, Fe, Zn, and B content using our LIBS system based on 104 PN samples from eight origins. More than 90 emission lines whose intensity values were above 1000 a.u. had their identities verified for PN samples. Univariate analysis based on sensitive emission lines could not meet the accuracy detection requirements of nutritive elements. For multivariable analysis, LS-SVM models based on the selected variables by Lasso weights for K, Ca, Mg, Zn, and B content detection obtained the best prediction performance with *R_p_* values of 0.9546, 0.9176, 0.9412, 0.9665, and 0.9569 and RMSEP of 0.7704 mg/g, 0.0712 mg/g, 0.1000 mg/g, 0.0012 mg/g, and 0.0008 mg/g, respectively. The Lasso model based on full spectra obtained the best result with and *R_p_* value of 0.9348 and an RMSEP of 0.0726 mg/g for Fe content detection. PLS models performed good both with full spectra and the selected variables by Lasso weights. The Lasso weight plots provided direct information about effective variable’s contribution to the quantitative analysis and can help researchers better understand the PN matrix.

LIBS technology combined with appropriate chemometric methods provided a fast, simple, and precise method for effective quantitative detection of nutrient elements in PN. However, further advances with more pharmaceutical analysis and other chemometric methods are still needed based on our study. Less and more effective variables should be discovered for the development of portable detection devices; ultimately, to provide a fast and accurate technique for laboratory analysis and online monitoring for Chinese herbal medicine quality and pharmaceutical analysis.

## Figures and Tables

**Figure 1 molecules-24-01525-f001:**
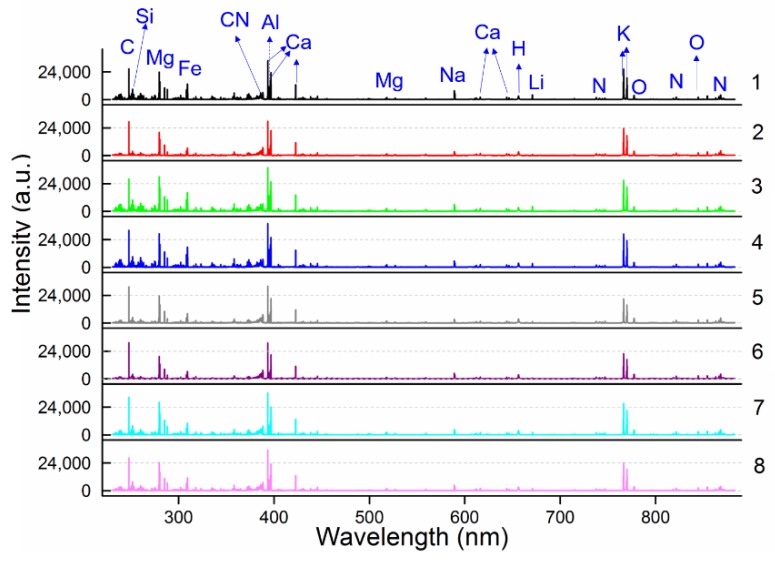
The average spectrum of each area in the range of 230.77–883.24 nm, 1: Xichou, 2: Yongde, 3: Malipo, 4: Mile, 5: Gejiu, 6: Gengma, 7: Shizong, 8: Qiubei.

**Figure 2 molecules-24-01525-f002:**
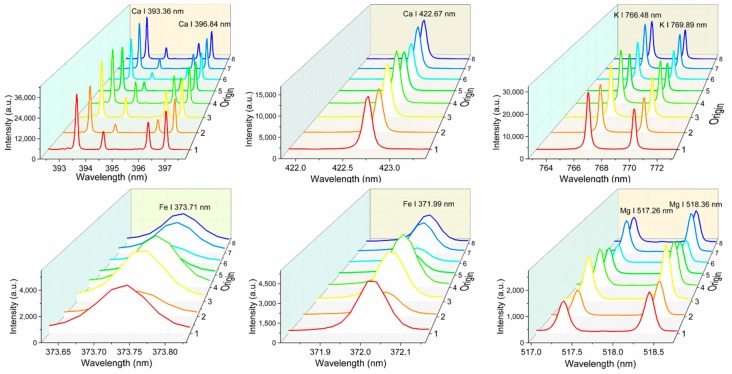
The emission lines of K, Ca, Fe and Mg for univariate analysis.

**Figure 3 molecules-24-01525-f003:**
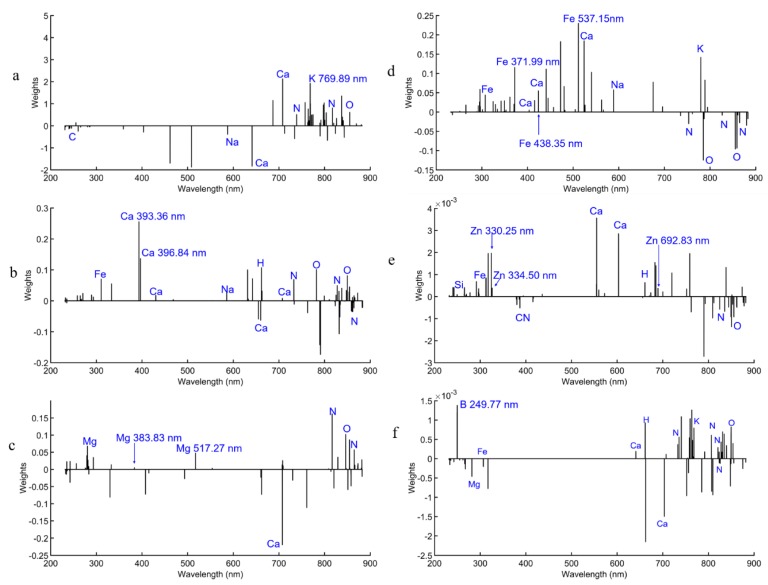
Weights plot of Lasso models for the nutrient elements K (**a**), Ca (**b**), Mg (**c**), Fe (**d**), Zn (**e**), and B (**f**).

**Figure 4 molecules-24-01525-f004:**
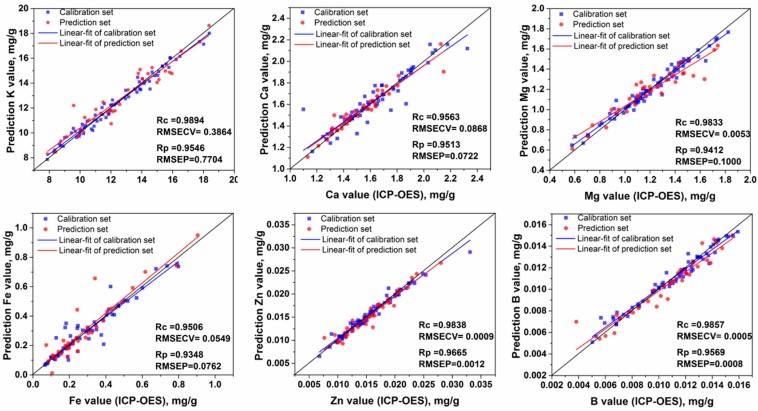
The best fitting plot of reference element content values and laser-induced breakdown spectroscopy (LIBS) measured element content values predicted by LS-SVM models (based on the selected variables) for K, Ca, Mg, and Zn and Lasso models for Fe (based on full spectra).

**Table 1 molecules-24-01525-t001:** Nutritive elements content (mg/g) of *Panax notoginseng*.

Element	Groups ^a^	1	2	3	4	5	6	7	8
	Number	13	13	13	13	13	13	13	13
K	Min	11.5682	8.2990	12.1523	11.0964	7.8709	6.4402	10.2981	8.7456
	Max	17.4899	15.4417	19.7162	18.3589	10.8974	13.8608	15.8312	18.0112
	Mean	14.0558	11.7255	16.0186	14.7329	9.3546	10.0683	12.8228	13.9661
	S.D.	1.6676	1.9894	2.5840	2.0317	0.9795	1.8793	1.4315	2.71071
Ca	Min	1.4194	1.1814	1.5225	1.2717	1.2216	1.0807	1.0997	1.4007
	Max	2.1659	1.8081	2.4316	2.3112	1.8509	2.3226	1.9342	2.3260
	Mean	1.7756	1.4200	1.9667	1.6736	1.4758	1.3450	1.4921	1.8509
	S.D.	0.2292	0.2062	0.4276	0.3137	0.1993	0.3683	0.2203	0.4987
Mg	Min	0.8821	0.8153	1.0908	1.1779	1.0813	0.5774	0.9143	1.1373
	Max	1.2918	1.4632	1.9072	1.6739	1.7435	1.1640	1.8240	1.8797
	Mean	1.1343	1.0583	1.5977	1.4081	1.3567	0.7984	1.2961	1.3825
	S.D.	0.1194	0.1824	0.2170	0.1559	0.2113	0.1802	0.2730	0.2240
Fe	Min	0.0288	0.0711	0.0837	0.1003	0.0661	0.0781	0.1154	0.0783
	Max	0.8145	0.3023	1.0317	0.9004	0.3862	0.5021	0.7329	0.7258
	Mean	0.2401	0.1598	0.5550	0.4918	0.1885	0.1903	0.3003	0.3785
	S.D.	0.0938	0.0668	0.1039	0.0546	0.0869	0.0779	0.0803	0.0840
Zn	Min	0.0147	0.0075	0.0122	0.0116	0.0103	0.0085	0.0069	0.0121
	Max	0.0351	0.0225	0.0303	0.0250	0.0159	0.0217	0.0159	0.0242
	Mean	0.0203	0.0130	0.0213	0.0192	0.0129	0.0131	0.0113	0.0183
	S.D.	0.0073	0.0037	0.0089	0.0044	0.0016	0.0041	0.0022	0.0041
B	Min	0.0091	0.0061	0.0057	0.0035	0.0038	0.0027	0.0074	0.0047
	Max	0.0154	0.0148	0.0165	0.0159	0.0134	0.0167	0.0147	0.0156
	Mean	0.0138	0.0105	0.0131	0.0130	0.0074	0.0079	0.0105	0.0132
	S.D.	0.0032	0.0024	0.0066	0.0057	0.0026	0.0036	0.0023	0.0038

^a^ 1: Xichou, 2: Yongde, 3: Malipo, 4: Mile, 5: Gejiu, 6: Gengma, 7: Shizong, 8: Qiubei.

**Table 2 molecules-24-01525-t002:** The obvious spectral emission lines of *Panax notoginseng* (PN) based on the NIST database.

Elements	Wavelength (nm)
C I	247.86, 296.72
Si I	250.68, 251.43, 251.61, 251.92, 252.41, 288.15
Fe I	302.06, 371.99, 385.99, 293.69, 498.24, 499.41
Fe II	253.54, 257.60, 259.37, 260.54, 263.08,
Mg I	277.98, 382.94, 383.23, 383.83, 389.19, 516.73, 517.27, 518.36
Mg II	279.55, 279.80, 280.27
Ca I	299.50, 300.09, 300.69, 422.67, 428.30, 428.94, 429.90, 430.25, 430.77, 431.87, 442.54, 443.50, 458.15, 458.60, 527.03, 558.87, 559.45, 559.85, 585.75, 610.27, 612.22, 616.22, 643.91, 644.98, 646.26, 649.38, 671.77, 714.82, 854.21
Ca II	315.89, 317.93, 373.69, 393.37, 396.85, 866.21
Sc II	364.37
CN	385.01 (CN 4-4), 385.44 (CN 3-3), 386.15 (CN 2-2), 387.12 (CN 1-1), 388.32 (CN 0-0)
Al I	394.40, 396.15
K I	693.87, 766.49, 769.90
Sr I	460.73
Sr II	407.77, 421.55
Na I	589.00, 589.59
H	656.28
O I	777.42, 844.67
Li I	670.79
N I	742.36, 744.23, 746.83, 818.48, 821.63, 824.23, 862.92, 868.02

**Table 3 molecules-24-01525-t003:** The obvious spectral emission lines of PN based on the NIST database.

Emission Lines nm	Calibration Set		Prediction Set	
	*R_c_*	RMSECV mg/g	*R_p_*	RMSEP mg/g
K I 766.49	0.8324	1.4002	0.7476	1.7601
K I 769.90	0.8413	1.3707	0.7836	1.6103
Ca II 393.37	0.6872	0.1284	0.6327	0.1282
Ca II 396.85	0.7764	0.1104	0.7118	0.1160
Ca I 422.67	0.7941	0.1062	0.7779	0.1074
Mg I 517.27	0.8403	0.1519	0.7564	0.1927
Mg I 518.36	0.7520	0.1840	0.7168	0.2025
Fe I 373.71	0.7509	0.1217	0.8378	0.1027
Fe I 371.99	0.8944	0.0820	0.8577	0.0973

**Table 4 molecules-24-01525-t004:** The results for multivariate analysis based on full spectra (22,036 variables) by PLS, LS-SVM, and least absolute shrinkage and selection operator (Lasso). RMSEP, root mean squared error of prediction.

Element	Model	Parameter	Calibration Set		Prediction Set	
			*R_c_*	RMSECV mg/g	*R_p_*	RMSEP mg/g
K	PLS ^d^	10 ^a^	0.9558	0.8120	0.9505	0.7152
	LS-SVM	(992.5, 799,024.9) ^b^	0.9800	0.3120	0.9391	0.8721
	Lasso	53 ^c^	0.9547	0.7740	0.9496	0.7956
Ca	PLS^d^	13 ^a^	0.9563	0.0868	0.9513	0.0722
	LS-SVM	(111.5, 16,929,970) ^b^	0.9799	0.0357	0.9135	0.1101
	Lasso	54 ^c^	0.9533	0.0872	0.9508	0.0798
Mg	PLS	11 ^a^	0.9270	0.1066	0.9171	0.1182
	LS-SVM	(236.1, 344,300.9) ^b^	0.9601	0.0986	0.9011	0.1246
	Lasso^d^	51 ^c^	0.9294	0.1022	0.9207	0.1110
Fe	PLS	4 ^a^	0.9234	0.0791	0.9334	0.0906
	LS-SVM	(311.9, 4,680,480) ^b^	0.9799	0.0451	0.9284	0.0854
	Lasso^d^	51 ^c^	0.9506	0.0549	0.9348	0.0762
Zn	PLS^d^	4 ^a^	0.9503	0.0017	0.9460	0.0016
	LS-SVM	(289.3, 1,593,133.6) ^b^	0.9886	0.0009	0.9060	0.0021
	Lasso	54 ^c^	0.9406	0.0015	0.9228	0.0019
B	PLS^d^	4 ^a^	0.9566	0.0008	0.9475	0.0010
	LS-SVM	(244.1, 48,672.5) ^b^	0.9866	0.0007	0.9036	0.0014
	Lasso	52 ^c^	0.9502	0.0009	0.9348	0.0009

^a^ is the parameter for PLS for the number of latent variables (LVs), ^b^ is the parameter for LS-SVM for penalty parameters (c) and kernel function parameters (g); ^c^ is the parameter for parameter for Lasso for the boundary value t; ^d^ means the best prediction model among three quantitative analysis methods for the specific element.

**Table 5 molecules-24-01525-t005:** The results for multivariate analysis based on the selected variables by PLS, LS-SVM, and Lasso.

Element (number)	Model	Parameter	Calibration Set		Prediction Set	
			*R_c_*	RMSECV mg/g	*R_p_*	RMSEP mg/g
K (64)	PLS	8 ^a^	0.9655	0.6852	0.9530	0.7853
	LS-SVM ^d^	(192.1, 6,274.1) ^b^	0.9894	0.3864	0.9546	0.7704
	Lasso	78 ^c^	0.9689	0.6491	0.9482	0.8239
Ca (73)	PLS	13 ^a^	0.9420	0.0638	0.9047	0.0757
	LS-SVM ^d^	(691.7, 19,536.2) ^b^	0.9890	0.0299	0.9176	0.0712
	Lasso	66 ^c^	0.9416	0.0639	0.9012	0.0776
Mg (61)	PLS	7 ^a^	0.9405	0.0957	0.9365	0.0979
	LS-SVM ^d^	(146.2, 3,195.7) ^b^	0.9833	0.0053	0.9412	0.1000
	Lasso	60 ^c^	0.9236	0.1080	0.9291	0.1034
Fe (66)	PLS ^d^	6 ^a^	0.9299	0.0684	0.9169	0.0724
	LS-SVM	(2585.9, 20,694.3) ^b^	0.9999	0.0002	0.9159	0.0891
	Lasso	100 ^c^	0.9070	0.0784	0.9034	0.0801
Zn (73)	PLS	6 ^a^	0.9158	0.0018	0.9613	0.0012
	LS-SVM ^d^	(81.3, 3,389.1) ^b^	0.9838	0.0009	0.9665	0.0012
	Lasso	100 ^c^	0.9561	0.0013	0.9100	0.0019
B (62)	PLS	6 ^a^	0.9579	0.0008	0.9432	0.0009
	LS-SVM ^d^	(569.8, 16,582.3) ^b^	0.9857	0.0005	0.9569	0.0008
	Lasso	70 ^c^	0.9515	0.0009	0.9195	0.0011

^a^ is the parameter for PLS for number of latent variables LVs, ^b^ is the parameter for LS-SVM for penalty parameters (c) and kernel function parameters (g); ^c^ is the parameter for Lasso for boundary value t; ^d^ means the best prediction model among three quantitative analysis methods for the specific element.

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
