# Peer review of "High-Sensitivity Determination of Nutrient Elements in Panax notoginseng by Laser-induced Breakdown Spectroscopy and Chemometric Methods"

_molecules, 2019, doi:10.3390/molecules24081525_

Round 1
Reviewer 1 Report
The authors present an interesting study on the use of laser induced breakdown spectroscopy (LIBS) for highly accurate and fast quantitative detection of 6 nutritive elements in PN samples from 8 producing areas. The study is sound and of interest to readers of Molecules. However, prior to potential acceptance for publication, the following minor changes are required:
1. Thorough English proofing is needed.
2. LIBS abbreviation not defined in the Introduction. (Re-check all abbreviations and make sure to define them throughout the text)
3. References don't require to cite the whole author name et al. (use only last name; e.g., Braga et al.).
4. What was the rationale for choosing PLSR, SVM and Lasso to do a comparison? This was not mentioned anywhere in the text. Wasn't Lasso-PLS also used? GA-PLS could be employed too as it was shown in a recent study: Žuvela and Liu. On feature selection for supervised learning problems involving high-dimensional analytical information. RSC Adv. 2016, 86, 82801-82809.
5. Section 2.4: What was the rationale for such training/validation set separation? Why not use a stratification method such as Kennard and Stone or duplex?
6. Section 2.4.2: The optimal nLVs is not the one that gives the minimum RMSECV; but the one that represents either the first minimum or the knee point of the RMSECV vs nLVs curve. Please re-do optimization of nLVs.
7. On Figure 4. Linear fit through predicted vs. measured is nonsense. It has no meaning whatsoever. Plot the y=x line instead. What is RMSECP?
Author Response
Point 1: - Thorough English proofing is needed.
Response 1: We checked the English grammar and spelling of the manuscript. The English of the manuscript was checked and enhanced carefully.
Point 2: - LIBS abbreviation not defined in the Introduction. (Re-check all abbreviations and make sure to define them throughout the text)
Response 2: Thank you for your suggestion, the complete spelling “Laser induced breakdown spectroscopy (LIBS)” was added in line 65 when the LIBS abbreviation first appeared in the Introduction. We also checked the full text for the same question.
Point 3: References don't require to cite the whole author name et al. (use only last name; e.g., Braga et al.).
Response 3: Thank you, we followed your advice to revise author name as Braga et al in line 69 and Carvalho et al. in line 71.
Point 4: What was the rationale for choosing PLSR, SVM and Lasso to do a comparison? This was not mentioned anywhere in the text. Wasn't Lasso-PLS also used? GA-PLS could be employed too as it was shown in a recent study: Žuvela and Liu. On feature selection for supervised learning problems involving high-dimensional analytical information. RSC Adv. 2016, 86, 82801-82809.
Response 4: PLS, SVM and Lasso were selected to establish quantitative models. The reason for choosing these three methods was mentioned in line 81-85 “PLS method proved to be establish robust model, mainly because PLS could reduce multicollinearity among variables and mitigate interference from matrix effect. Other studies have also confirmed SVM and Lasso are also good methods to build excellent regression models”. The rationales of PLS, SVM and Lasso were introduced in section 2.4.2. Multivariate analysis methods.
Genetic algorithms (GA) is a common and good method to filter feature variables in spectral data processing. In this study, LIBS data is large and complex which has 22015 variables in one spectrum. It takes a long time to use the GA to train the LIBS data by the ordinary computer. And the filtered variables change with the different algorithm parameters in GA operation. Lasso is a good dimensionality reduction method with short calculation time and stable selected variable. So, this study used Lasso algorithm for regression and feature variable screening as an attempt to apply Lasso in LIBS research.
Point 5: - Section 2.4: What was the rationale for such training/validation set separation? Why not use a stratification method such as Kennard and Stone or duplex?
Response 5: Before quantitative analysis, 104 samples were partitioned into calibration set (72 samples) and prediction set (32 samples) based on Kennard-Stone (KS) algorithm which could avoid bias in sample selection. The KS algorithm were added in line 146.
Point 6: Section 2.4.2: The optimal nLVs is not the one that gives the minimum RMSECV; but the one that represents either the first minimum or the knee point of the RMSECV vs nLVs curve. Please re-do optimization of nLVs.
Response 6: Thank you for your suggestion and what you pointed out is correct. We re-examined the LVs and found that the LVs were consistent with the first the knee point being the minimum. And this point was added for LVs selection criteria in line 156-158: The five-fold cross-validation procedure was used and the number of LVs was selected when the first minimum or the knee point for the root mean squared error of calibration (RMSECV) vs LVs curve was obtained.
Point 7: On Figure 4. Linear fit through predicted vs. measured is nonsense. It has no meaning whatsoever. Plot the y=x line instead. What is RMSECP?
Response 7: Thank you. Figure 4 was added dashed lines of y=x. Linear fit plots were shown in Figure 4 and reveal the relationship between common chemical method ICP-OES (measured element values) and the novel fast detection technology LIBS (predicted element values). The closer the correlation coefficient is to 1, the closer detection accuracy of LIBS technology is to the conventional technology.
The RMSECP in the figure is wrong and has been corrected to RMSEP. RMSEP is root mean square error of prediction set.

Reviewer 2 Report
This study might be useful for the rapid determination of elements in Chinese Herbs. The total experimental setup is good, and the expression is logic and fluent, and some minor revisions is needed: 1. When preparing Panax notoginseng samples, how to confirm the representative of the samples? i.e., does heterogneity exist in each sample? 2. in spectral recording, while 4 × 4 × 5? what is approximate size of the crater, or what is the interval distance of each hit? 3. The prediction set might be better to be changed as validation set. 4. Some tables are not easily understood, such as Table 4,Table 5 what is the paprameter1? 5. The unit of microelement, such as Fe, Zn, B, might be suitable as mg/kg rather than g/kg.Author Response
Response to Reviewer 2 Comments
Point 1: When preparing Panax notoginseng samples, how to confirm the representative of the samples? i.e., does heterogneity exist in each sample?
Response 1: The Panax notoginseng samples was prepared by experts from Yunnan University of Traditional Chinese Medicine who engaged in long-term research on Panax notoginseng. According to the origin characteristics of Yunnan Province, we have chosen representative sources of Panax notoginseng. The samples were preserved independently and without crossover. From the experts' point of view, it is possible to ensure that there are no sample with large differences in the same origin; No outliers are found from spectral curves of the same origin; From the results of elemental measurement by ICP-OES in Table 1, elements content of the samples from the same origin is in the normal range with no outliers. So, there is no heterogneity exist in each sample.
Point 2: in spectral recording, while 4 × 4 × 5? what is approximate size of the crater, or what is the interval distance of each hit?
Response 2: The 4 × 4 craters obtained after laser ablation and each crater was formed by 5 times accumulation of laser pulses. In this regard, it has already been explained in Line 125-126 of the manuscript. The interval distance of each hit is 2mm and surface area of the crater is approximate 0.3 mm2.
Point 3: The prediction set might be better to be changed as validation set.
Response 3: Cross-validation is a common method used in machine learning to build models and validate model parameters. The validation set is a set used to verify model parameters in cross-validation. The prediction set is a sample set that is predicted based on the optimal training model obtained by cross-validation. In order to distinguish from the validation set in cross-validation, it is not appropriate to change prediction set to validation set.
Point 4: Some tables are not easily understood, such as Table 4,Table 5 what is the paprameter1?
Response 4: “parameter1” in Table 4 and Table 5 was changed as “parameter” which is the parameter of the models. “a” was the parameter for PLS is number of latent variables LVs, “b” was the parameter for LS-SVM is penalty parameters (c) and kernel function parameters (g); “c” was the parameter for parameter for Lasso is boundary value t. The related explanation exists below Table 4 and Table 5 in the manuscript.
Point 5: The unit of microelement, such as Fe, Zn, B, might be suitable as mg/kg rather than g/kg.
Response 5: The content of microelement (Fe, Zn, B) and K, Ca and Mg in Panax notoginseng is measured by the same instrument ICP-OES. In order to retain the same effective number, the units of all element contents are expressed in the same expression unit mg/g in the manuscript.
